# Veno-Arterial-Venous Extracorporeal Membrane Oxygenation in a Critically Ill Patient with Coronavirus Disease 2019

**DOI:** 10.3390/medicina56100510

**Published:** 2020-09-30

**Authors:** Joung Hun Byun, Dong Hoon Kang, Jong Woo Kim, Sung Hwan Kim, Seong Ho Moon, Jun Ho Yang, Jae Jun Jung, Oh-Hyun Cho, Sun In Hong, Byung-Han Ryu, Hyun Oh Park, Jun Young Choi, In Seok Jang, Jong Duk Kim, Chung Eun Lee

**Affiliations:** 1Department of Thoracic and Cardiovascular Surgery, College of Medicine, Gyeongsang National University, Gyeongsang National University Changwon Hospital, Changwon 51472, Korea; jhunikr@naver.com (J.H.B.); drk82@hanmail.net (D.H.K.); cs99kjw@hanmail.net (J.W.K.); clariboy@naver.com (S.H.K.); hoya_m@naver.com (S.H.M.); junhoyah@hanmail.net (J.H.Y.); thoracoscope@gmail.com (J.J.J.); 2Department of Internal Medicine, College of Medicine, Gyeongsang National University, Gyeongsang National University Changwon Hospital, Changwon 51472, Korea; zenmd@naver.com (O.-H.C.); hsun0702@hanmail.net (S.I.H.); qudhany@naver.com (B.-H.R.); 3Department of Thoracic and Cardiovascular Surgery, Institute Health Science, College of Medicine, Gyeongsang National University, Gyeongsang National University Jinju Hospital, Jinju 52727, Korea; romejuliet@naver.com (H.O.P.); jychoi@gnu.ac.kr (J.Y.C.); isjang@gnu.ac.kr (I.S.J.); frogeye1@hanmail.net (J.D.K.)

**Keywords:** COVID-19, respiratory distress syndrome, extracorporeal membrane oxygenation

## Abstract

Patients with cardiopulmonary failure may not be fully supported with typical configurations of extracorporeal membrane oxygenation (ECMO), either veno-arterial (VA) or veno-venous (VV). Veno-arterial-venous (VAV)-ECMO is a technique used to support the cardiopulmonary systems during periods of inadequate gas exchange and perfusion. In the severe case of coronavirus disease 2019 (COVID-19), which simultaneously affects the heart and lung, VAV-ECMO may improve a patient’s recovery potential. We report the case of a 72-year-old woman with acute respiratory distress syndrome and circulatory failure following COVID-19, who was treated with VAV-ECMO.

## 1. Introduction

In December 2019, cases of pneumonia of an unknown etiology, now known as coronavirus disease 2019 (COVID-19), spread rapidly around the world. Clinical manifestations of COVID-19 include myocarditis and acute respiratory distress syndrome (ARDS) in severe cases [1]. To our knowledge, there are no definitive reports about extracorporeal membrane oxygenation (ECMO) mode for treating severe ARDS and left ventricular dysfunction due to COVID-19. Such patients may be successfully treated with veno-arterial-venous-ECMO (VAV-ECMO) by concurrently supporting both the heart and lungs. Here, we report the case of patient with severe ARDS and left ventricular dysfunction due to COVID-19. We applied VAV-ECMO to support both the heart and lungs. Since then, the patient has been successfully weaned from both ECMO and ventilator and was discharged without complications.

## 2. Case Reports

The patient was a 72-year-old woman whose condition deteriorated six days after confirmation of COVID-19. She had a past medical history of hypertension, an implanted pacemaker, and was on hydrocortisone for secondary adrenal insufficiency. Multifocal pneumonia was detected on chest X-ray (Figure 1A). The arterial blood gas analysis (ABGA) from the right radial artery revealed the following: pH, 7.53; PaCO_2_, 30 mmHg; and PaO_2_, 72 mmHg at a ventilator setting of inspired oxygen fraction (FiO_2_) 50%, tidal volume (TV) 8 mL/kg, and positive end-expiratory pressure (PEEP) 8 cmH_2_O. Transthoracic echocardiography (TTE) revealed ejection fraction (EF) of 55%. Tazobactam, hydrocortisone, hydroxychloroquine, lopinavir/ritonavir, and trimethoprim/sulfamethoxazole were administered concurrently. However, after five days, at a ventilator setting of FiO_2_ 90%, TV 8 mL/kg, PEEP 10 cmH_2_O, the patient’s PaO_2_/FiO_2_ deteriorated to 78, peak inspiratory pressure (PIP) increased to 27 mmHg, and lung infiltration worsened (Figure 1B). Despite administration of norepinephrine, the patient’s hemodynamics was unstable, oliguria developed, and TTE revealed an EF of 45%. In this situation, once the prone position was considered, since this patient showed a sharp drop in arterial pressure (70/40 mmHg) and bradycardia (40/min) when the position was changed, we decided to apply VAV-ECMO since we believed that protecting the lungs and ensuring optimal perfusion to other organs were necessary.

We inserted a 24-French gauge (Fr) drainage cannula (Edwards Lifescience LLC, Irvine, CA, USA) via left common femoral vein, a 22-Fr venous return cannula via the right common femoral vein, and an 18-Fr arterial return cannula (Edwards Lifescience LLC, Irvine, CA, USA) via the right femoral artery. The divided return flow was monitored using an ultrasonic flow sensor (ELSA, Transonic Systems, Ithaca, NY, USA) and controlled by partially clamping the venous return cannula. The arterial return flow was maintained at about 30–40% of the cardiac output, and the venous return flow was maintained at about 80–90% of the cardiac output.

The patient’s PaO_2_/FiO_2_ was improved to 400 with a ventilator setting of FiO_2_ 30%, TV 4 mL/kg, and PEEP 7 cmH_2_O, and PIP remained at 19–20 cmH_2_O. The patient’s hemodynamics was stabilized without norepinephrine. The lung infiltration was the most exacerbated on the second day after initiating VAV-ECMO (Figure 1C). On the tenth day of VAV-ECMO support, chest radiography (Figure 1D) showed improvement. There was also a decrease in pronounced levels of troponin I and lactate 10 days after ECMO treatment (Table 1). We were able to wean her from VAV-ECMO according to the VAV-ECMO weaning protocol defined by our department (Figure 2). We performed ventricular setting during 10 days after ECMO treatment (Table 1). A percutaneous dilatational tracheostomy was performed, and the patient was weaned off the ventilator support six days after ECMO removal. Figure 1E shows chest radiograph after removing the ventilator support.

The patient was confirmed to be negative for the COVID-19 virus three times by a real-time reverse transcription polymerase chain reaction (rRT-PCR) test. She was discharged without complications. We used hydrocortisone for about three weeks during ECMO operation from the hospitalization period. After that, we stopped it and patient was discharged after about eight weeks.

## 3. Discussion

In December 2019, cases of pneumonia of an unknown etiology, now known as COVID-19, were reported in Wuhan, Hubei province, China [1]. Since then, COVID-19 has spread rapidly around the world, and the World Health Organization (WHO) has declared it a global pandemic. Huang et al. presented a list of clinical manifestations, including fever, cough, and dyspnea, as well as radiographic evidence of pneumonia and organ dysfunction in severe cases [2]. In a study of 150 patients from two hospitals in Wuhan, China, among the 68 fatal cases, 36 patients (53%) died of respiratory failure, 5 (7%) with myocardial damage died of circulatory failure, and 22 (33%) died of both [3]. Although definitive treatment guidelines are yet to be determined, the WHO has assembled a set of interim guidelines which recommend applying veno-venous (VV)-ECMO to patients with COVID-19-related ARDS [4]. It is also believed that there is a need to respond quickly to circulatory failure. We think the initial indicators of circulatory failure are easy to miss, and irreversible organ damage is likely to occur. If, in a situation where respiratory failure is more severe and circulatory failure is thought to be modifiable by medications (inotropic agents), clinicians may consider applying VV-ECMO. If circulatory failure is more severe, many clinicians may consider applying veno-arterial (VA)-ECMO with mechanical ventilator care. However, depending on the patient’s clinical situation, optimal circulatory support or pulmonary protection may not be sufficiently achieved because these are combined in a series of clinical situations. According to Extracorporeal Life Support Organization (ELSO) guidelines, the application of VV-ECMO is suggested when the risk of mortality is 80% or greater. In terms of respiratory medicine, an 80% mortality risk is associated with a PaO_2_/FiO_2_ < 100 on FiO_2_ >90% and/or a Murray score of 3–4. The application of VA-ECMO is indicated when inadequate tissue perfusion has manifested as hypotension and low cardiac output despite adequate intravascular volume [5]. Bartlett et al. [6] reported ECMO is considered when the situation presenting PaO_2_/FiO_2_ <80 lasts longer than 6 h. Our patient suffered from PaO_2_/FiO_2_ <78 on FiO_2_ 290% for 12 h and had a Murray score of 3.5. The patient developed oliguria and her BP decreased below 90/60 mmHg despite the administration of norepinephrine. If only veno-arterial (VA)-ECMO was applied, differential hypoxia, which is the severe complication of VA-ECMO, might occur due to poor function of the lungs and desaturated blood from the left ventricle could cause cerebral and myocardial hypoxia [7]. One of the major challenges is to decide on how to treat the injured lungs to keep alive and to promote the healing. The potential options range from complete lung rest [8]. If VV-ECMO was applied alone, it would not provide direct hemodynamic support. In this regard, VAV-ECMO primarily protects the lung and other organs, which can contribute to improving the patient’s recovery.

## 4. Conclusions

There are several opinions on how to approach the treatment of the novel COVID-19, and a definitive recommendation needs to be established in the future. In the interim, although not a definitive solution, we believe that VAV-ECMO may be appropriate for treating severe cases of COVID-19.

## Figures and Tables

**Figure 1 medicina-56-00510-f001:**
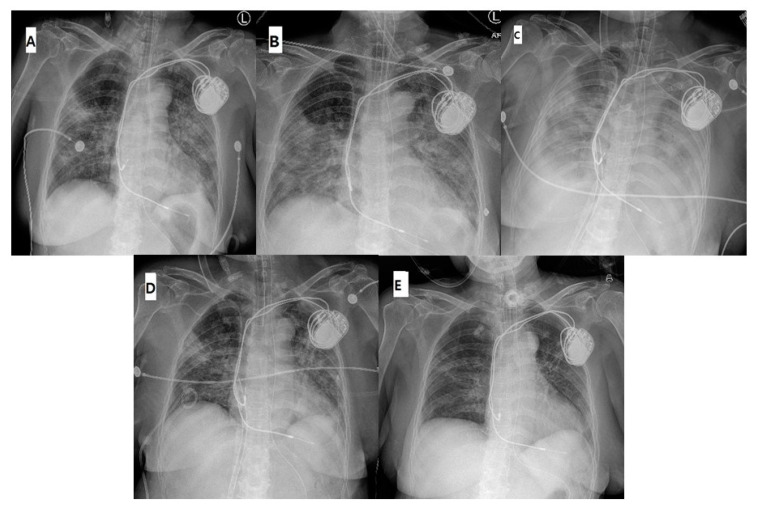
(**A**) Initial chest radiograph showing multifocal pneumonia; (**B**) at the time of VAV-ECMO application, infiltration was worsened; (**C**) on the second day after VAV-ECMO application, multifocal pneumonia was most exacerbated; (**D**) on the day of withdrawal of VAV-ECMO support; and (**E**) at the time of removal of the ventilator.

**Figure 2 medicina-56-00510-f002:**
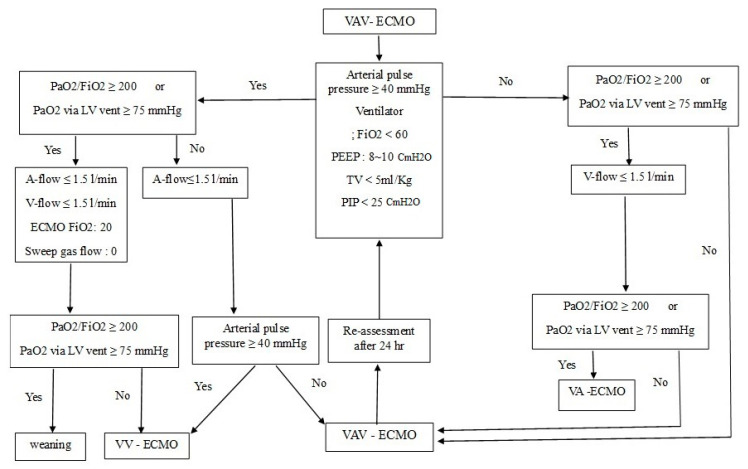
Proposed weaning algorithm for VAV-ECMO of our department.

**Table 1 medicina-56-00510-t001:** The changes of clinical parameters during 10 days after ECMO treatment.

Clinical Parameters	Day 1	Day 3	Day 5	Day 10
PaO_2_/FiO_2_ on ventilator	78	375	230	200
Peak Inspiratory Pressure (cmH2O)	33	22	21	25
Cardiac Troponin I (ng/L)	77	50	21	10
Lactate (mmol/L)	3.4	2.1	1.5	0.8
Left ventricular ejection fraction (%)	45	-	48	58

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
