# Peer review of "Veno-Arterial-Venous Extracorporeal Membrane Oxygenation in a Critically Ill Patient with Coronavirus Disease 2019"

_medicina, 2020, doi:10.3390/medicina56100510_

Round 1

Reviewer 1 Report

Here the authors present a case report of a COVID-19 patient treated with V-VA ECMO. The case report is interesting, but it requires improvement.

1) Of paramount importance is to provide a Table with some of the data (ventilator setting, BGA, hemodynamics and laboratory) at intervals: I would suggest baseline, 1, 3, 5, 10 days. Without it is difficult to call it a "case report". As a reader I need to see visually the time course in a more detailed fashion, the verbal description you provide is not detailed enough.

2) Hemodynamics data are largely missing: invasive pressure monitoring, echo findings, vascular resistances and, most importantly, cardiac output. You report that the arterial line was kept at 40% of the cardiac output. How much was it? In my experience COVID-19 patients have a high cardiac output.

3) ECMO settings throughout the days are very important

4) Do you have a CT scan instead of the x ray?

5) Important: I believe that lactates are relevant in this context. Deciding to apply the hemodynamic support implies that perfusion was inadequate. How did you assess it? 

6) Did you perform LV unloading? 

7) Was the hydrocortisone administration continued throughout? I would discuss the point in relationship with the RECOVERY trial

8) I would cite one of the most recent reviews on the topic, I suggest Quintel et al. Anesthesiology 2020.

9) The problem of myocardial injury is serious in COVID-19: do you have markers of myocardial damage? Is it possible to see the time course of it?

Overall, I think that many more details are required.

Author Response

Response to Reviewer 1’ comments

1) Of paramount importance is to provide a Table with some of the data (ventilator setting, BGA, hemodynamics and laboratory) at intervals: I would suggest baseline, 1, 3, 5, 10 days. Without it is difficult to call it a "case report". As a reader I need to see visually the time course in a more detailed fashion, the verbal description you provide is not detailed enough.

We thoroughly appreciate the reviewer’s valuable comment. Thus, during 10 days after ECMO treatment, we incorporated a table 1 with some data including ventricular setting, PIP, cardiac troponin I, lactate, and LV EF.

Table 1. The changes of clinical parameters during 10 days after ECMO treatment.

Clinical parameters

Day 1

Day 3

Day 5

Day 10

PaO2/FiO2 on ventilator

78

375

230

200

Peak Inspiratory Pressure (cmH2O)

33

22

21

25

Cardiac Troponin I (ng/L)

77

50

21

10

Lactate (mmol/L)

3.4

2.1

1.5

0.8

Left ventricular ejection fraction (%)

45

-

48

58

2) Hemodynamics data are largely missing: invasive pressure monitoring, echo findings, vascular resistances and, most importantly, cardiac output. You report that the arterial line was kept at 40% of the cardiac output. How much was it? In my experience COVID-19 patients have a high cardiac output.

We appreciate the reviewer’s thoughtful comment. As the reviewer suggested, we added LVEF in table 1. ECMO flow of arterial line: 1.5 ~2 L/min. If the patient had undergone severe septic shock, a high flow of ECMO would have been necessary. However, the patient was not in severe septic condition, and the arterial blood pressure was maintained at about 130/70 mmHg when norepinephrine administration was stopped at an ECMO flow of arterial line of 1.5 ~ 2 L/min under moderate LV dysfunction.

3) ECMO settings throughout the days are very important

As shown in table 1, during 10 days after ECMO treatment, we provided ECMO setting according to the protocol in figure 2.

ECMO A line flow : 1.5~2 L/min. ECMO V line flow :4~5 L/min. ECMO fiO2 : Starting at initial 1.0, weaning was performed at 0.2. Sweep gas : Starting from the initial 4 L, it decreased according to the Pco2 level (35 ~40 mmHg) on the ABGA. ACT : It was maintained for 180-250 sec.

4) Do you have a CT scan instead of the x ray?

Because of the possibility of spread of virus, we have not analyzed the CT scan.

5) Important: I believe that lactates are relevant in this context. Deciding to apply the hemodynamic support implies that perfusion was inadequate. How did you assess it?

We absolutely agree with your opinion. We regard lactate level as a very important marker of inadequate tissue perfusion and urine output as an important marker. The blood lactate levels were added to the table 1.

6) Did you perform LV unloading?

We did not perform this. However, we use the transaortic catheter venting technique for LV unloading. In our center, indication for left heart decompression include narrow pulse pressure, LVEF < 25 %, failure of AV opening in TTE, and persistent pulmonary edema despite adequate volume control. Our patient was not included in the criteria above.

7) Was the hydrocortisone administration continued throughout? I would discuss the point in relationship with the RECOVERY trial

We added sentences as follows; We used it for about 3 weeks during ECMO operation from the hospitalization period. After that, we stopped it and patient was discharged after about 8 weeks.

8) I would cite one of the most recent reviews on the topic, I suggest Quintel et al. Anesthesiology 2020.

As the reviewer suggested, we added revised sentences and reference as follows; One of the major challenges is to decide on how to treat the injured lungs to keep alive and to promote the healing. The potential options range from complete lung rest [8].

9) The problem of myocardial injury is serious in COVID-19: do you have markers of myocardial damage? Is it possible to see the time course of it?

We measured cardiac-specific troponin I during 10 days and added its levels to table 1.

Reviewer 2 Report

Thank you for the opportunity to review this work.

The authors describe a case using VAV ECMO to support a 72-year-old woman with ARDs from COVID-19.The patient survived.

This statement "there are no definitive reports about ECMO mode for treating severe ARDS" is partially correct. There are very few "definitive" reports about ECMO in COVID-19, but there are certainly many reports that have now been published. See PubMed ID: 32911986, 32906085, 32798468, 32780089 just to list a few.

"Case Repots" is misspelled on page 2. 

72 years of age is a relative contraindication for most centers that provide ECMO. This patient also had an extensive cardiac history. While the outcome was good, this is a highly questionable patient that most centers would not put on ECMO.

The ABG and ventilator settings did not support initiation of VAV. An ejection fraction of 45% (without any inotropic agent) should have been more than adequate. The P/F ratio was poor, but it is not stated whether the patient was proned or if other adjuncts were used. The PIP doesn't support ECMO. The CXR seems to only be worse once "lung rest" settings were applied, on ECMO.

It is not clear how a PEEP of 7 was chosen.

"The" percutaneous tracheostomy should be written as "a percutaneous"

The references used are mostly from Wuhan and are outdated. The ELSO guidelines for COVID-19 are note cited (Bartlett RH, Ogino MT, Brodie D, et al. Initial ELSO Guidance Document: ECMO for COVID-19 Patients with Severe Cardiopulmonary Failure [published correction appears in ASAIO J. 2020 Aug;66(8):e113]. ASAIO J. 2020;66(5):472-474).

The discussion is disjointed. This sentence "In contrast, clinicians who consider veno-arterial (VA)-ECMO paired with mechanical ventilator care will be common" makes no sense and needs to be rewritten.

Author Response

Response to Reviewer 2’ comments

1) This statement "there are no definitive reports about ECMO mode for treating severe ARDS" is partially correct. There are very few "definitive" reports about ECMO in COVID-19, but there are certainly many reports that have now been published. See PubMed ID: 32911986, 32906085, 32798468, 32780089 just to list a few.

What we are talking about is not that there is no definitive report on whether ECMO is effective in severe ARDS caused by COVID 19, but that there is no clear report on which ECMO mode is effective. We have experienced about 700 cases of ECMO care in the last 10 years. All currently known ECMO modes were implemented. However, we think ECMO care for severe ARDS with cardiomyopathy is difficult and requires much discussion. In my experience, what matters in ARDS from other causes, including COVID 19, will be how aggressive lung resting will be, and by what criteria the treatment for stressed cardiomyopathy will be. I don't think you can support heart if you simply perform VV ECMO for lung support and show unstable hemodynamics after using inotropics. At what point ECMO will be implemented for heart and lung support and which ECMO mode is best to use, we will have to think deeply and that is the purpose of this report. We are working on VAV ECMO and trying to establish indication & weaning criteria.

2) "Case Repots" is misspelled on page 2.

We corrected this typo.

3) 72 years of age is a relative contraindication for most centers that provide ECMO. This patient also had an extensive cardiac history. While the outcome was good, this is a highly questionable patient that most centers would not put on ECMO.

We partially agree with your opinion. We don't give much of the meaning of the word aging. If the aspect of efficient use of medical resources is largely considered, it is appropriate to apply ECMO to patients who are younger and have a higher survival probability. However, we believe that it is appropriate to give patients a chance to survive, even if they are elderly and have many underlying diseases, unless it is an option of circumstances due to a lack of ECMO. In particular, the guardian of this patient also requested active treatment from us. In our center, even if the patient is octogenarian, it is judged that the damage of the heart or lung is reversible, and if it is thought that there is a possibility of returning to daily life even if it is not a long period after recovery, active treatment is provided.

4) The ABG and ventilator settings did not support initiation of VAV. An ejection fraction of 45% (without any inotropic agent) should have been more than adequate. The P/F ratio was poor, but it is not stated whether the patient was proned or if other adjuncts were used. The PIP doesn't support ECMO. The CXR seems to only be worse once "lung rest" settings were applied, on ECMO.

We do not consider only ABGA and ventilator settings when applying V-AV ECMO. We have a lot of experience with ECMO and have relatively clear indication for ECMO application. Of course, refer to the ELSO guide line clearly. However, this is not considered an absolute criterion. There are standards for applying different ECMO modes, and in particular, different papers are currently being written regarding the standards for applying VAV ECMO. As mentioned in the case presentation section, despite the high dose of norepinephrine in the patient, arterial blood pressure was maintained around 90/60 mmHg, and oliguria occurred without significant change even after volume replacement. In TTE, EF decreased to 45%. Of course, it may be questioned whether the ejection fraction of 45% is indeed the standard for ECMO application, but patients with an EF of 55% suddenly decreased to 45%, and systemic resistance did not increase despite injecting inotropics. If not exerted, obvious shock symptoms occur. This will put the patient into a vicious cycle. As mentioned in discussion section, according to Extracorporeal Life Support Organization (ELSO) guidelines, the application of VV-ECMO is suggested when the risk of mortality is 80% or greater. In terms of respiratory medicine, an 80% mortality risk is associated with a PaO2/FiO2 < 100 on FiO2> 90% and/or a Murray score of 3-4. (ELSO Guidelines for Cardiopulmonary Extracorporeal Life Support. Extracorporeal Life Support Organization, Version 1.4, August 2017. Ann Arbor, MI, USA. (https://www.elso.org/Resources/Guidelines.aspx).

Our patient had PaO2 / FiO2 of 78 and a Murray score of 3.5. Once the prone position was considered, we judged that this patient should apply ECMO rather than prone position because the arterial pressure rapidly decreased (60/40 mmHg) and bradycardia (40/min) when the position was changed.

As you pointed out, we revised sentence as follows; In this situation, once the prone position was considered, since this patient showed a sharp drop in arterial pressure (70/40 mmHg) and bradycardia (40/min) when the position was changed, we decided to apply VAV-ECMO since we believed that the need to protect the lungs and ensure optimal perfusion to other organs were necessary.

We applied VAV ECMO to patients as we thought that circulation support as well as lung protection was important. And, as you know, applying ECMO does not immediately improve the patient's lung or heart. As the patient's heart or lung deteriorates slowly, it is common to recover slowly over time.

5) It is not clear how a PEEP of 7 was chosen.

In our center, we usually think that low tidal volume and low PIP are important, so for lung protection, we aim for PEEP less than 8 cmH2O, initial TV less than 50%, FiO2 less than 40%, and PIP less than 25 cmH2O. Then in right radial ABGA we try to keep PaO2 above 70 mmHg.

6) "The" percutaneous tracheostomy should be written as "a percutaneous"

We corrected it.

7) The references used are mostly from Wuhan and are outdated. The ELSO guidelines for COVID-19 are note cited (Bartlett RH, Ogino MT, Brodie D, et al. Initial ELSO Guidance Document: ECMO for COVID-19 Patients with Severe Cardiopulmonary Failure [published correction appears in ASAIO J. 2020 Aug;66(8):e113]. ASAIO J. 2020;66(5):472-474).

At that time, few cases of VAV ECMO were applied to COVID 19 patients, and almost all died. So the reference seems to be outdated. We revised sentences as follows; The application of VA-ECMO is indicated when inadequate tissue perfusion has manifested as hypotension and low cardiac output despite adequate intravascular volume. Bartlett et al. reported ECMO is considered when the situation presented PaO2/FiO2 <80 lasts longer than 6 h. Our patient suffered from PaO2/FiO2 <78 on FiO290% for 12 hours and had a Murray score of 3.5.

8) The discussion is disjointed. This sentence "In contrast, clinicians who consider veno-arterial (VA)-ECMO paired with mechanical ventilator care will be common" makes no sense and needs to be rewritten.

As the reviewer commented, we revised it as follows; If circulatory failure is more severe, many clinicians may consider applying VA-ECMO with mechanical ventilator care.

Round 2

Reviewer 1 Report

Thanks for addressing my concerns, I think that the paper improved a lot.